# Effect of Silica Sodalite Functionalization and PVA Coating on Performance of Sodalite Infused PSF Membrane during Treatment of Acid Mine Drainage

**DOI:** 10.3390/membranes11050315

**Published:** 2021-04-26

**Authors:** Nobuhle C. Ntshangase, Olawumi O. Sadare, Michael O. Daramola

**Affiliations:** 1Faculty of Engineering and Built Environment, School of Chemical and Metallurgical Engineering, University of the Witwatersrand, Wits, Johannesburg 2050, South Africa; 876791@students.wits.ac.za; 2Department of Chemical Engineering, Faculty of Engineering, Built Environment and Information Technology, University of Pretoria, Hatfield, Pretoria 0028, South Africa; wumisadare@gmail.com

**Keywords:** acid mine drainage, heavy metals, silica sodalite, functionalization, membranes, poly vinyl alcohol, polysulfone

## Abstract

In this study, silica sodalite (SSOD) nanoparticles were synthesized by topotactic conversion and functionalized using HNO_3_/H_2_SO_4_ (1:3). The SSOD and functionalized SSOD (fSSOD) nanoparticles were infused into a Polysulfone (Psf) membrane to produce mixed matrix membranes. The membranes were fabricated via the phase inversion method. The membranes and the nanoparticles were characterized using Scanning Electron Microscopy (SEM) to check the morphology of the nanoparticles and the membranes and Fourier Transform Infrared to check the surface chemistry of the nanoparticles and the membranes. Thermal stability of the nanoparticles and the membranes was evaluated using Themogravimetry analysis (TGA) and the degree of hydrophilicity of the membranes was checked via contact angle measurements. The mechanical strength of the membranes and their surface nature (roughness) were checked using a nanotensile instrument and Atomic Force Microscopy (AFM), respectively. The textural property of the nanoparticles were checked by conducting N_2_ physisorption experiments on the nanoparticles at 77 K. AMD-treatment performance of the fabricated membranes was evaluated in a dead-end filtration cell using a synthetic acid mine drainage (AMD) solution prepared by dissolving a known amount of MgCl_2_, MnCl_2_·4H_2_O, Na_2_SO_4_, Al(NO_3_)_3_, Fe(NO_3_)_3_·9H_2_O, and Ca_2_OH_2_ in deionized water. Results from the N_2_ physisorption experiments on the nanoparticles at 77 K showed a reduction in surface area and increase in pore diameter of the nanoparticles after functionalization. Performance of the membranes during AMD treatment shows that, at 4 bar, a 10% fSSOD/Psf membrane displayed improved heavy metal rejection >50% for all heavy metals considered, expect the SSOD-loaded membrane that showed a rejection <13% (except for Al^3+^ 89%). In addition, coating the membranes with a PVA layer improved the antifouling property of the membranes. The effects of multiple PVA coating and behaviour of the membranes during real AMD are not reported in this study, these should be investigated in a future study. Therefore, the newly developed functionalized SSOD infused Psf membranes could find applications in the treatment of AMD or for the removal of heavy metals from wastewater.

## 1. Introduction

Global industrialization, in particular in developing countries, is causing a serious effect on the environment. For instance, heavy metals are naturally released to the environment through rock weathering and volcanic eruptions. At times, they are mostly discharged by human activities like mining, agriculture, and other industries, resulting in the discharge of heavy metals to the environment and underground water [1]. This poses a danger to humans, animals, the environment, and aquatic life when the heavy metals concentration limits are exceeded [2]. Heavy metals deposition is also dangerous because they are non-biodegradable and tend to accumulate on living organisms as they can be taken up by plants and be translocated into edible parts of the plant [2,3]. Due to these detrimental impacts on human beings, plants, and animals, numerous studies have explored the removal of heavy metals from water bodies using membranes [4,5], agricultural waste [6,7], and other methods.

Several methods have been investigated for the removal of heavy metals, these include adsorption, chemical precipitation, electrostatic interaction, ion exchange, and membrane filtration [2,3,8]. Adsorption is widely studied in literature for the removal of heavy metals like Fe^3+^, Mn^2+^, Cu^2+^, and Zn^2+^ using agricultural wastes [9] and natural zeolite [10]. This is because adsorption has high efficiency and a relatively low-cost of operation. In addition, it is simple and easy to operate, and is neutral to harmful compounds and toxic pollutants [3,11]. In the chemical precipitation process, lime and limestone are usually used as precipitation agents to precipitate the heavy metals and form an insoluble heavy metal hydroxide [12]. This method is favourable because of the availability of precipitant agent, low cost of operation, process simplicity, and its effectiveness in the removal of heavy metals from a solution with a high concentration (>1000 mg/L) [12]. Nevertheless, its application is limited because it produces large amount of sludge, requires a large amount of chemicals, and a long process time for remediation wastewater that contains heavy metals. Hence, it is mostly applicable in abandoned mines [13].

Electrostatic interaction involves the exchange of current between a cathode plate and an insoluble anode through a metal-bearing aqueous solution, but its drawback is the corrosion of the electrodes, and hence a high cost of maintenance [12]. Ion exchange involving the exchange of cations and anions between the heavy metal and the exchanger solid has also been widely used. Its application is limited by its ineffectiveness at high metal concentrations and low pH levels, and it is non-selective in the presence of multiple metals [12]. Attention of many researchers have been drawn towards the application of membranes for treatment of heavy metals in acid mine drainage. This is due to their high selectivity, low pressure requirements, and small space of operation. Nevertheless, application of polymer membranes to the removal of heavy metals from wastewater is adversely affected by membrane fouling, which increases its operational cost [5,14]. Hence, inorganic nanoparticles have been introduced into organic polymer membranes creating mixed matrix membranes (MMMs) to limit the fouling mechanism [15,16]. Filler nanoparticles investigated for heavy metal removal include, but are not limited to, carbon nanotubes [1,17,18] and hydroxy sodalite (HSOD) [5]. Zeolites are known for their high surface area, uniform pore size, well defined microporosity, and high hydrothermal stability [19,20]. A type of zeolite, hydroxy sodalite (HSOD), was evaluated by Daramola et al. (2015) in a polymer-infused membrane. The fabricated and tested membrane displayed good permeability but with limited selectivity in the treatment of AMD. The trade-off between membrane permeability and selectivity is a major concern in membrane application. Therefore, the porous SSOD nanoparticles embedded in the membrane are expected to produce a high flux and reduced selectivity [21]. 

Functionalization improves the surface properties of nanoparticles (specific surface area, pore-size distribution, or pore volume), increases the presence of functional groups, and enhances their structural stability [2]. The modification of nanoparticles can influence the charge of the membrane, smooth the surface, and enhance the hydrophilicity of the membrane, hence limiting fouling but enhancing performance [14]. The functionality of the functional groups in a material is determined by the type of functionalization method. For example, the cocondensation method is performed by adding the functionalization agent during the synthesis process, whereas in the grafting method the material is synthesized and calcined to remove surfactant and then the functionalization agent is added [22]. The grafting method is characterized by a nonuniform distribution of functional groups, high loading of functional groups, structure maintenance of the material, and exterior positioning of the functional groups when compared to the co-condensation method [22,23,24]. 

Furthermore, membrane permeability is influenced by its hydrophilicity as it influences the transportation of water through the membrane. Polysulfone is known to be a hydrophobic polymer, making it more susceptible to fouling. In spite of this, its chemical resistance to harsh environments and its ease of modification to improve its performance [25] have favoured the application of polysulfone. Fouling in membrane is defined as a deposition of unwanted particles on the surface or in the pores of the membrane, hence resulting in a decline of the membrane performance [14]. Poly vinyl alcohol (PVA) has been explored in membrane coating for improvement of wettability and mechanical strength of membranes [26,27]. 

Against this background, this study therefore investigated the synthesis of silica sodalite nanoparticles by topotactic conversion and the application of the particles in a membrane application. In addition, the effect of functionalization by the grafting method on the dispersion of the SSOD nanoparticles in the polysulfone polymer matrix and its performance evaluation for the treatment of AMD are studied. Furthermore, the effect of Poly vinyl alcohol (PVA) on the hydrophilicity of the fabricated SSOD and fSSOD-infused Psf membranes was investigated.

## 2. Materials and Methods

### 2.1. Materials

Tetraethyl orthosilicate (TEOS) (C_8_H_20_O_4_Si) (reagent grade 98%) and tetramethylammonium hydroxide (TMAOH) (C_4_H_13_NO) solution (25 wt.% in H_2_O) were used in the synthesis of RUB-15. These chemicals were purchased from Sigma Aldrich (Pty), Johannesburg, South Africa. Propionic acid, purchased from CC Imelmann Pty Ltd. (Johannesburg, South Africa), was used for RUB-15 pre-treatment to form SSOD. H_2_SO_4_ and HNO_3_ (38%) were used for functionalization. Polysulfone (average Mw = 22,000 g/mol) beads were used as the polymer backbone; N,N-dimethylacetamide (purity ≥ 99.9% (Mw = 87.12 g/mol)) was used as the solvent; Poly vinyl alcohol (87–90% hydrolysed (average Mw = 30,000–70,000)); and Maleic acid (purity ≥ 99%) was used as the cross-linking agent. All these chemicals, except propionic acid, were purchased from Sigma Aldrich (Pty) (Merck), Johannesburg, South Africa, and used with no further purification. Synthetic AMD was prepared in-house by dissolving MgCl_2_, MnCl_2_·4H_2_O, Na_2_SO_4_, Al(NO_3_)_3_, Fe(NO_3_)_3_·9H_2_O, and Ca_2_OH_2_ in deionized water. 

### 2.2. Synthesis and Fabrication Procedures

#### 2.2.1. Nanoparticles Synthesis

SSOD was synthesized by following a procedure described elsewhere [21,28,29]. 18.5 g TEOS was mixed with 32 mL TMAOH and stirred at 500 rpm overnight to form a homogenous mixture. The solution was then autoclaved at 140 °C for seven days. The precipitate was then centrifuged and rinsed with acetone. The particles were dried at 60 °C for 24 h, producing RUB-15. An amount of 0.1 g of the RUB-15 particles was treated with 30 mL of 5 M propionic acid. This solution was magnetically stirred at 500 rpm for 180 min. Particles were recovered by centrifuging, and were then washed with distilled water to a neutral pH. Afterwards, the washed particles were then dried in an oven at 60 °C for 24 h. The treated RUB-15 particles were then calcined at 600 °C for 3 h to obtain silica sodalite nanoparticles.

#### 2.2.2. Functionalization of Silica Sodalite Nanoparticles

The surface of the synthesized SSOD nanoparticles was functionalized using carboxylation protocol reported elsewhere [30] in order to enhance their dispersion in the polymer matrix. An amount of 1 g of SSOD was dispersed in a mixture of 75 mL H_2_SO_4_ (97%) and 25 mL HNO_3_ (38%) and stirred overnight at 40 °C to obtain fSSOD. The fSSOD nanoparticles were recovered by centrifugation and were repeatedly washed with deionized water to a neutral pH, and dried in an oven at 100 °C for 24 h.

#### 2.2.3. Fabrication of Mixed Matrix Membrane 

The nanofiltration membrane was fabricated by dissolving 10 g polysulfone in 50 mL of N,N-dimethylacetamide. Varying amounts of the sodalite nanoparticles were added to the mixture to make a loading of 5 wt.% and 10 wt.%. This mixture was ultrasonicated for 10 min to uniformly disperse the nanoparticles and stirred on a magnetic stirrer for 24 h to form a homogenous solution. The homogenous solution was further ultrasonicated to remove bubbles from the solution. The solution was hand-cast on a glass-plate using “doctor blade” and left for 10 s to allow the solvent to evaporate. The cast membrane was then immersed in a bath of deionized water, and the membrane was obtained by phase inversion as per the method for the fabrication of nanofiltration membrane. The obtained membranes were soaked in deionized water overnight to remove impurities, after which they were oven-dried at 60 °C.

#### 2.2.4. Membrane Coating

The fabricated membranes were coated by following a procedure reported elsewhere [31]. An amount of 1 g of PVA was dissolved in 100 mL deionised water and stirred at 90 °C to form a coating solution. The cross-linking solution was prepared by dissolving 1 g maleic acid in deionized water to a total volume of 100 mL. The membranes were immersed in the PVA solution for 3 min after which they were cross-linked by immersing into maleic acid solution for a further 3 min. The membranes were then dried in an oven at 60 °C overnight.

#### 2.2.5. Preparation of Synthetic AMD Solution

Synthetic AMD was prepared by dissolving a known amount of MgCl_2_, MnCl_2_·4H_2_O, Na_2_SO_4_, Al(NO_3_)_3_, Fe(NO_3_)_3_·9H_2_O, and Ca_2_OH_2_ in deionized water to obtain the various concentration indicated in Table 1. The pH of the solution was adjusted by adding 0.1 M H_2_SO_4_ and 0.1 M NaOH to achieve a pH of 2.8.

### 2.3. Characterization of Nanoparticles and Membranes

SEM images of the produced SSOD and fSSOD nanoparticles and membranes were obtained by using ZEISS scanning electron microscopy (SEM) (Carl Zeiss NTS Ltd., Oberkochen, Germany) at an accelerating voltage of 20 kV. The Micrometrics Tristar 3000 (RS232) was used for Brunauer-Emmett-Teller (BET) ((Micromeritics Instrument Corp., 4356 Communications Drive, Norcross, GA, USA) analysis to determine the surface area, pore volume, and pore diameter of the nanoparticles. Crystalline structure of the synthesized nanoparticles was analysed using Bruker XRD (Bruker, Kalsruhe, Germany) D2 Phaser with a CuKα target at a wavelength of 1.54 Å, a tube voltage of 30 kV and a tube current of 10 mA. The samples were scanned at a step size of 0.026° and a rate of 8.5°/min from 5°–90° of 2θ. The Miller indices (hkl) were obtained from the reference library of Diffrac.Eva software. The presence of functional groups on the nanoparticles was checked by using PerkinElmer FTIR spectrometer equipped with a high-performance deuterated triglycine sulphate (DTGS) detector and KBr beam splitter (PerkinElmer Inc, Waltham, MA, USA). The thermal behaviour of nanoparticles was determined by TA instrument SDT Q600 simultaneous DSC/TGA analyser (TA instrument, New Castle, DE, USA). The heating rate was 20 °C/min from room temperature to 800 °C and with a nitrogen gas flow of 50 mL/min. The mechanical properties of the membranes were evaluated using TA.XT plus texture analyser at a speed of 8.6 mm/s and ambient temperature (Stable Micro System, Surrey, UK). The contact angle of the membranes was measured using the sessile drop method (OCA 15 EC GOP, Data physics). Deionized water was used as a probe liquid dispensed at 1 μL/s. 

### 2.4. Membrane Performance Evaluation

Membrane performance was evaluated by conducting permeation experiments using a dead-end filtration cell with a 400 mL holding volume (purchased from Memcon SA (Pty)). The filtration cell has a 10 bar max working pressure, and an allowable working temperature in the range 5–80 °C. A section of the synthesized membrane was cut and used in the filtration cell. The air-tightness of the membrane in the cell was ensured using two rubber seals. The used membrane had a diameter of 7.2 cm and an effective permeation area of 32 cm^2^. To ensure a steady state prior to the acquisition of flux data, the membrane was compacted overnight in the filtration cell with deionized water.

Permeate volume was collected at an interval of 1 h at pressure between 3–5 bar and the membrane flux was calculated using Equation (1). As a way of investigating the fouling behaviour of the membrane, filtration experiments were carried out at a fixed pressure of 4 bar for a duration of 3 h and the membrane flux was evaluated at a 30 min interval. The membrane rejection was calculated using Equation (2):(1)$$J_{p}=\frac{V_{p}}{A}$$
(2)$$R_{i}\left(\%\right)=\frac{C_{fi}-C_{pi}}{C_{fi}}\times 100$$
where $J_{p}$ is the water flux$\left(\frac{L}{h\cdot m^{2}}\right)$; *V_p_* is the permeate volumetric flow rate (L/h); A is the effective area of the membrane in cm^2^ (32 cm^2^); R_i_ is the membrane rejection (expressed in percentage) for the metals; and $C_{fi}$ and $C_{pi}$ are the concentration (in mg/L) of dissolved metal, i, in the feed and in the permeate, respectively (mg/L). The concentration of the dissolved metal in the feed and in the permeate was determined using an Atomic Absorption Spectroscopy.

The flux recovery ratio (FRR), which indicates the ability of the membrane to recover from fouling, was obtained using Equation (3). A better antifouling property is indicated by a higher FRR [3]. This indicates that there is a low reduction in flux over time, and it might be easy to restore the membrane flux with cleaning.
(3)$$FRR\;\left(\%\right)=\frac{J_{w2}}{J_{w1}}\times 100$$
where *J*_*w*1_ and *J*_*w*2_ are the initial flux and flux after a cleaning process, respectively. 

## 3. Results and Discussion

### 3.1. Nanoparticles Characterization

The surface morphology of SSOD and fSSOD is depicted in Figure 1a,b, respectively. The SEM image of fSSOD showed that functionalization of SSOD preserved the sheet-like morphology of the nanoparticles. A smoothing of the exterior surface of fSSOD can be observed, indicating reduced roughness [32]. This is a characteristic of the grafting method as it is said to maintain the structure of a material [22,23,24]. The BET surface area was obtained as 200 cm^2^/g and 188 cm^2^/g, while the pore volume was obtained as 0.242 cm^2^/g and 0.240 cm^3^/g for SSOD and fSSOD nanoparticles, respectively. A reduction in surface area and increase of pore diameter after functionalization was reported in the literature for carbon nanotubes [33]. This is caused by the blockage of pore entrances due to the formation of functional groups on the surface of the nanoparticles [33].

The XRD pattern in Figure 2 shows that sodalite was successfully formed as all the required peaks appear. This pattern was identified by the DIFFRAC EVA software as silica sodalite, dehydrated with a chemical formula (C_2_H_7_NO)(Si_6_O_12_), similar SSOD patterns have been obtained in the literature [21,29,30]. However, the diffraction intensities increased, confirming the modification with additional functional groups [34]. 

FTIR spectra for the SSOD and fSSOD nanoparticles are presented in Figure 3. The fingerprint region (400–1400cm^−1^) of the FTIR spectra shows the presence of all the characteristic functional groups peaks of SSOD. The increase in peak definition in this region for fSSOD is said to be an indication of quantitative increment of hydrate oxide on the SSOD surface as a result of the deformation of O-H bonding and combination of C-O (979 and 1273 cm^−1^) stretching in surface aromatic carboxylic acids [34]. The covalent bond at 1080 cm^−1^ in SSOD and fSSOD confirms the existence of a dense silica network where the oxygen atoms play the role of bridges between each two silicon sites; these are asymmetric stretching vibrations of T-O-T (T = Si/O) [35]. The absorption peaks observed at 1273 cm^−1^ and 979 cm^−1^ are attributed to C-O stretching characteristic of carboxylic acid groups and alcohol groups, respectively. Absorption peaks located at 1728 cm^−1^ can be assigned to C=O stretching from carboxylic acid groups and overlapped with C=C band from aromatic rings [32]. There are a few new groups on the fSSOD spectrum indicated in Figure 2, but the expected functional group -COOH is not visible. 

Thermal stability of the as-produced nanoparticles was examined by the thermo gravimetric analyser (TGA) as shown in Figure 4. A 5% reduction in weight of the nanoparticles in the region <100 °C can be associated with removal of content water in the sodalite. Above this temperature, there is a further 1.4% weight loss for SSOD associated with the decomposition of interlayer propionic acid in the SSOD particles [27]. The 3.54% weight loss above 100 °C in fSSOD is associated with the decomposition of the -COOH functional group [36,37]. The increased weight loss for fSSOD is a confirmation of the presence of additional functional groups [30]. These observations are also reported in the literature [28,30]. These results show that the as-produced nanoparticles are stable at higher temperatures as there is very little weight loss.

### 3.2. Membrane Characterization

SEM images in Figure 5 evaluate the impact of nanoparticles distribution on the membranes. Figure 5a–c depict the surface morphology of pure psf (0% SSOD/Psf) at lower magnification 10% SSOD/Psf membranes, and pure psf at higher magnification, respectively. Figure 5a shows that the surface of the membrane is dense with no visible pores. However, the pores were more visible and evenly distributed when the magnification was increased. Moreover, the surface of 10% SSOD/Psf in Figure 5b is dense with no clear visible pores, attributable to the suppression of micro-void formation caused by high concentration of SSOD particles.

Figure 6a shows the 10% SSOD/Psf membrane with an area marked A showing agglomeration of nanoparticles on the cross sectional area of the membranes. This has been shown to limit membrane performance due to the increased free void volume [38]. The 10% fSSOD/Psf membrane in Figure 6b shows an improved distribution of nanoparticles on the membrane surface. This can be attributed to the functionalization of nanoparticles that have been proven to improve the dispersion of nanoparticles on polymer matrix and hence improve membrane performance [37]. There is no visible separate layers on the 10% fSSOD/Psf/PVA membrane as shown in Figure 6c. This shows that there was a good bond formation between the Psf and PVA layers. The results obtained in this study are consistent with what was obtained in the literature [38,39].

Mechanical strength of the fabricated membranes is depicted in Figure 7. It can be observed that infusing nanoparticles on the polymer matrix improves the membrane’s Young Modulus from 50 MPa for Psf to 79 MPa and 82 MPa for 10% fSSOD/Psf and 10% SSOD/Psf, respectively. A maximum Young Modulus of 138 MPa in 10% SSOD/Psf/PVA and tensile strength of 19.7 kPa in 10% fSSOD/Psf/PVA was obtained after the membranes were coated with PVA. 

The effect of nanoparticle loading on hydrophilicity of the membrane was investigated by measuring the contact angle and surface roughness of the membrane. The results are presented in Figure 8. Psf is a known hydrophobic polymer, as shown in Figure 8, with a high contact angle (88°). There is relatively no impact on the membranes hydrophilicity when SSOD was added. The high contact angle implies that SSOD is also hydrophobic [40]. However, as could be seen in Figure 8a, the infusion of fSSOD nanoparticles (10% SSOD/Psf) into the Psf improved the hydrophilicity of the pure polymer (75° as compared to 88°). Functionalization is known to improve the surface properties of nanoparticles such as the specific surface area, pore-size distribution, or pore volume. An increase in the presence of functional groups has been shown to enhance structural stability of membranes [2]. The literature reveals that infusing functionalized nanoparticles into membranes increases their hydrophilicity because the functionalized nanoparticles tend to travel towards the surface of the membrane, thus inducing the hydrophilic nature of the surface [41]. This could be because of modification of nanoparticles that influence the charge of the membrane, smooth the surface, and enhance the hydrophilicity of the membrane. In addition, embedding functionalized particles limits fouling and enhances the membrane performance [14]. Furthermore, the 0% SSOD/Psf (pure polysulfone membrane) showed the highest average roughness of 16.38 nm and root mean square roughness (Rms) of 13.06 nm. This could be attributed to the hydrophobic nature of pure polysulfone. The decrease in roughness has been observed with the incorporation of nanoparticles. However, 10% fSSOD/Psf membrane showed the higher average roughness compared to 10% SSOD/Psf, which could be due to the alteration in the structure of the membrane layer because of the functionalized nanoparticles embedded in the polymer membrane. The contact angle of 10% SSOD/Psf/PVA and 10% fSSOD/Psf/PVA was reduced by 10% and 17%, respectively, when compared to Psf. This can be attributed to the presence of the hydrophilic PVA layer [26].

The surface roughness of the membranes was determined by AFM. The root mean square (R_ms_), which is the standard deviation of all the vertical distances within the enclosed area, and the average roughness (R_a_), which indicates the mean roughness of the surface relative to the centre plane, are presented in Figure 9b. The fSSOD-loaded membranes show increased surface roughness when compared to the SSOD-loaded membranes (see Figure 9). This was not expected because the infusion of functionalized nanoparticles should reduce the surface roughness of the membrane because of the added carboxylic acid groups on the nanoparticles surface that can reduce the deposition of foulants on the surface of the membrane [38]. The unexpected reduction in surface roughness of the 10% SSOD/Psf membrane could be attributed to the ineffective dispersion of nanoparticles within the membrane. Nevertheless, adding the PVA coating on the membrane improved the membrane roughness leading to a reduction in the fouling rate of the membrane. 

### 3.3. Membrane Performance Evaluation

#### 3.3.1. Effect of Functionalization of Nanoparticles on Membrane Performance

The synthesized SSOD and fSSOD nanoparticles were infused into Psf to obtain 10% SSOD/Psf membrane and 10% fSSOD/Psf membrane. First, their separation performance was compared through pure water permeation at varying membrane pressures and then in the treatment of AMD. Figure 10 depicts the pure water permeation behaviour of these membranes while Figure 11 shows their performance during the treatment of AMD. A general trend of increasing pure water flux at increasing pressure was observed for all the membranes (see Figure 10). This can be attributed to the increased driving force that facilitated the enhancement of the capillary pressure of the membrane resulting in enhanced water permeation through the membranes [38]. The infusion of nanoparticles has improved the permeability of the membranes. Consequently, a maximum membrane flux of 2.2 L/m^2^·h for 10% fSSOD/Psf membrane and 2.3 L/m^2^·h for 10% SSOD/Psf membrane was at 5 bar. This behaviour could be attributed to the presence of nanoparticles within the membrane, and the opening of the fractional free volume of the membrane at increasing transmembrane pressure [42]. Functionalizing the SSOD nanoparticles to fSSOD had negligence effect on the membrane permeability. In a study conducted by Rameetse et al. (2020), the functionalization process resulted in an improved permeability. This was attributed to hydrogen bonding between water molecules influenced by oxygen elements from the added functional groups, thus forming a thin hydration layer on the membrane surface. It is likely that the small amounts of functional groups do not have a noticeable effect, hence an increase in the time of functionalisation to increase the number of functional groups is recommended.

Although the SSOD-loaded membrane showed high pure water flux, a poor selectivity can be observed for the 10% SSOD/Psf membrane in Figure 11. This trade-off between selectivity and permeability of the membrane can be attributed to the poor dispersion of nanoparticles resulting in the formation of agglomerates. The agglomerates result in increased macro voids, hence, allowing large heavy metals to pass through [38]. Functionalisation of SSOD resulted in enhanced rejection in the range of 51.5–74.2%. The general observed rejection trend is Fe^3+^ > Ca^2+^ >Al^3+^ >Mn^2+^ >Mg^2+^ > Na^2+^. The improved rejection can be attributed to the successful functionalization. Pure Psf membrane showed 55.2% and 49.5% rejection for mg^2+^ and mn^2+^, respectively, when compared to the 10% SSOD-loaded membrane having 13.3% and 9.1% for mg^2+^ and mn^2+^ rejections, respectively. The improved rejection performance of the pure psf membrane could be attributed to its hydrophobic nature resulting in the membrane having lower flux compared to the SSOD loaded membranes (see Figure 10). The higher flux exhibited by 10% SSOD might have contributed to its poor rejection performance. Moreover, the concentrations of mg^2+^ and mn^2+^ metals cations are lower compared to other metals as shown in Table 1. This could also be responsible for their rejection by the pure Psf membrane.

#### 3.3.2. Effect of Functionalization and PVA Coating on Fouling Behaviour

The reduction of membrane performance due to fouling is one of the challenges in membrane applications. Figure 12 shows the membranes flux over a period of time. A reduction in flux can be observed for all the evaluated membranes. The 10% SSOD/Psf membrane displayed the highest reduction of about 90.8% in membrane flux while the PVA-coated membrane displayed about 68.2% reduction in membrane flux when both membranes were evaluated under similar conditions. The drastic flux reduction in the 10% SSOD/Psf membrane (90.8%) and that of the Psf membrane (87.8%) can be attributed to the hydrophobic nature of the two membranes. Fouling is more pronounced on hydrophobic membranes as the hydrophobic foulants molecules are mostly driven to the membrane surface [43,44]. The 10% fSSOD/Psf membrane shows an improved fouling resistance when compared to that of the 10% SSOD/Psf membrane. This is likely due to the improved dispersion of nanoparticles due to functionalization; hence, most of the membrane surface had the active nanoparticles. The reduction in membrane flux of the PVA-coated membrane when compared to that of the non-PVA-coated membrane could be attributed to the increase in membrane thickness due to the coating. However, the PVA-coated membranes showed a higher fouling resistance when compared to that of the non-PVA-coated membranes. 

Fouling is classified as organic, inorganic or biofouling which may be reversible or irreversible depending on the type of foulants [45]. According to Arnal et al. (2011), the extent of fouling in membranes can be minimised by applying feed pretreatment, suitable operation conditions, and reasonable membrane cleaning methods. Nevertheless, the feed pretreatment and optimisation of the operating conditions do not eliminate the problem of membrane fouling. Therefore, to increase the life span of a membrane, cleaning methods have to be applied to reverse the fouling should the fouling be a reversible fouling. Figure 13 shows the flux recovery ratio of the PVA-coated membranes after 30 min of cleaning with deionized water. The 10% fSSOD/Psf/PVA membrane was able to recover its initial flux by more than 80% while the 10% SSOD/Psf/PVA membrane was only able to recover about 55% of its initial flux after use. The reduced recovery rate on the 10% SSOD/Psf/PVA could be attributed to the possible agglomerates that could have formed due to poor dispersion. The agglomerates are said to form defects that could entrap the foulants molecules, thus, the cleaning process may not be effective [46]. The 81% FRR obtained for 10% fSSOD/Psf/PVA is less than the 98% FRR obtained in a reported study [47]. This could be attributed to the different nanoparticles and coating material used in this study as compared to the reported study. 

## 4. Conclusions

Silica Sodalite (SSOD) and functionalized silica sodalite (fSSOD) nanoparticles were successfully synthesized and infused into a Psf membrane. To enhance the fouling resistance of these membranes, the membranes were coated with a PVA layer. Performance of the fabricated membranes was tested using pure water permeation and during the treatment of synthetic AMD. The 10% fSSOD/Psf showed an improved permeability and selectivity with heavy metal rejection >50% during the AMD treatment. For the PVA-coated membranes, a flux recovery ratio (FRR) of 81% and 56% was obtained for 10% fSSOD/Psf/PVA membrane and 10% SSOD/Psf/PVA membrane, respectively, after a cleaning process. The obtained results indicate that functionalization is able to improve dispersion of nanoparticles in mixed matrix membranes and it could enhance membrane selectivity. In addition, coating the membrane with a PVA layer enhanced the fouling property of the membranes, especially the membranes embedded with fSSOD. Though the effect of multiple PVA coating on the performance of these membranes is not a focus of this study, it is recommended that future studies should consider this. In addition, operating stability and membrane reproducibility should be investigated to ascertain the integrity of these membranes during AMD treatment. At the same time, it is important to evaluate the performance of these membranes using real AMD because presence of other impurities in the real AMD might greatly affect the performance of the membranes reported in this study. In spite of the aforementioned shortcomings, this study has reported novel functionalized SSOD infused Psf membranes that could find applications in the treatment of AMD or for the removal of heavy metals from wastewater. 

## Figures and Tables

**Figure 1 membranes-11-00315-f001:**
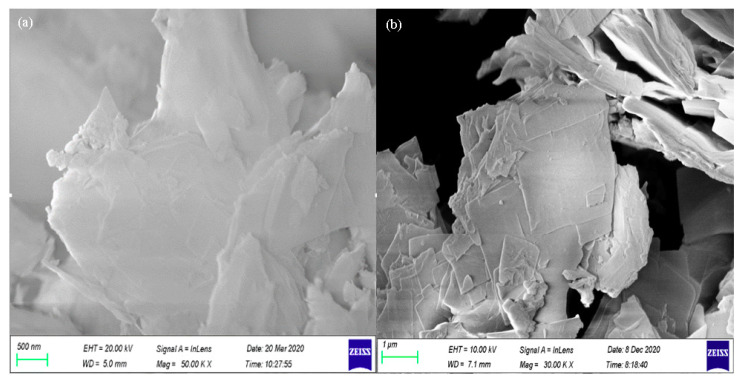
SEM images of (**a**) SSOD and (**b**) fSSOD.

**Figure 2 membranes-11-00315-f002:**
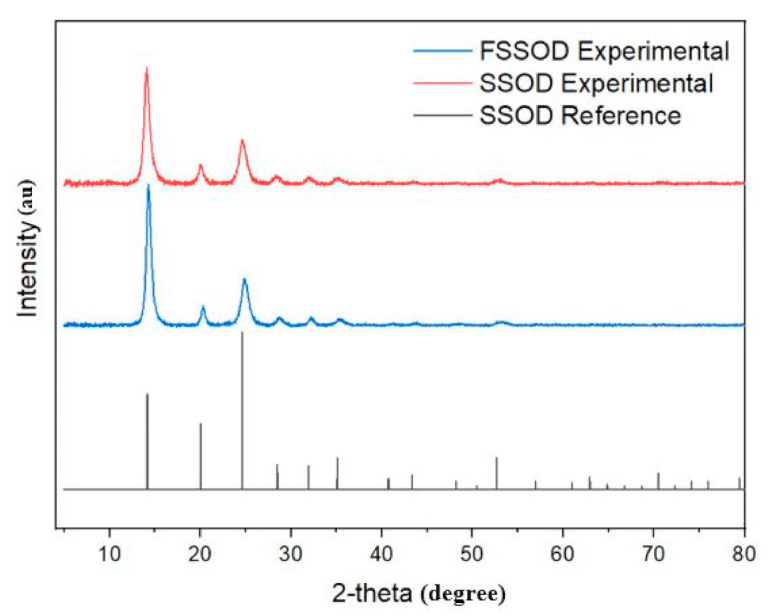
XRD pattern of SSOD and fSSOD compared to the XRD pattern from DIFFRAC EVA database.

**Figure 3 membranes-11-00315-f003:**
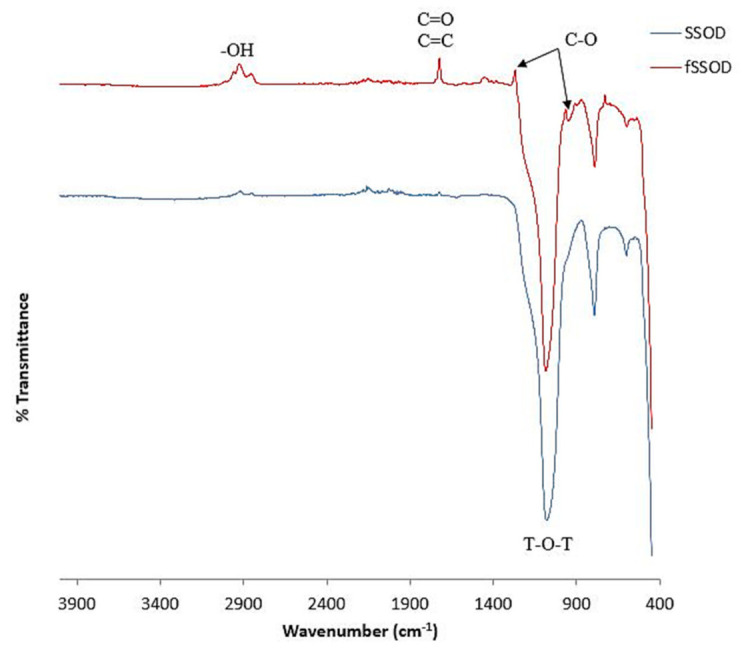
FTIR spectra of SSOD and fSSOD nanoparticles.

**Figure 4 membranes-11-00315-f004:**
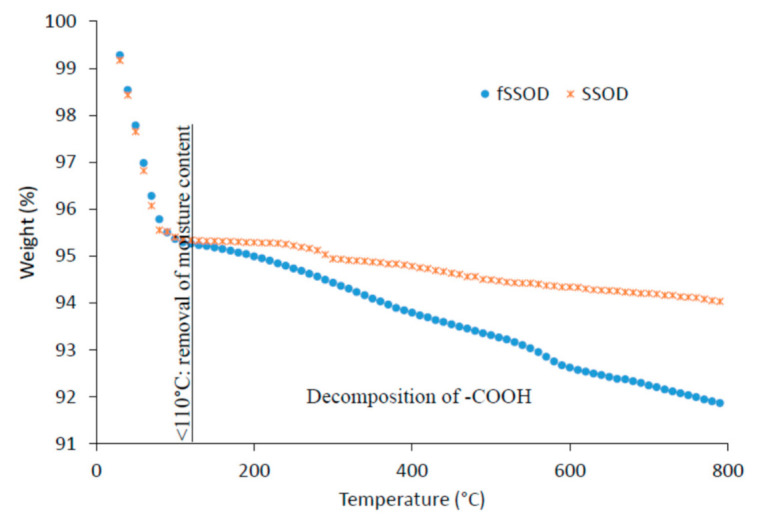
TGA profile of fSSOD and SSOD.

**Figure 5 membranes-11-00315-f005:**
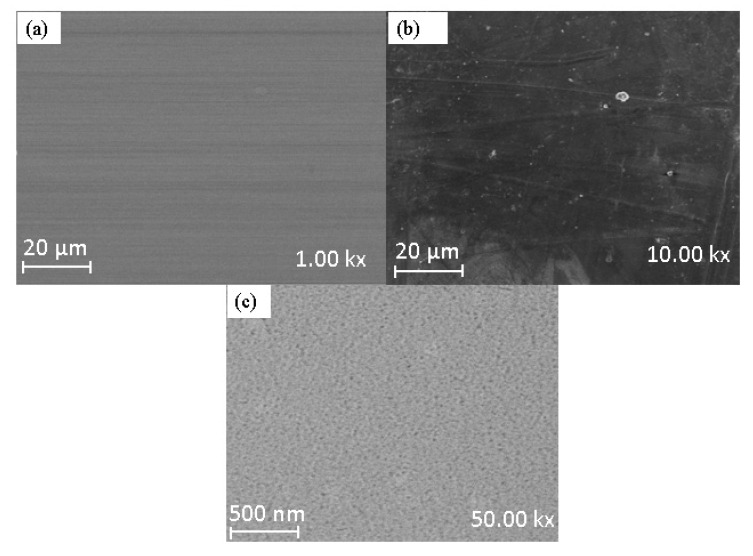
Surface morphology of (**a**) 0% SSOD/psf at lower magnification, (**b**) 10% SSOD/psf, membranes, and (**c**) 0% SSOD/psf at higher magnification.

**Figure 6 membranes-11-00315-f006:**
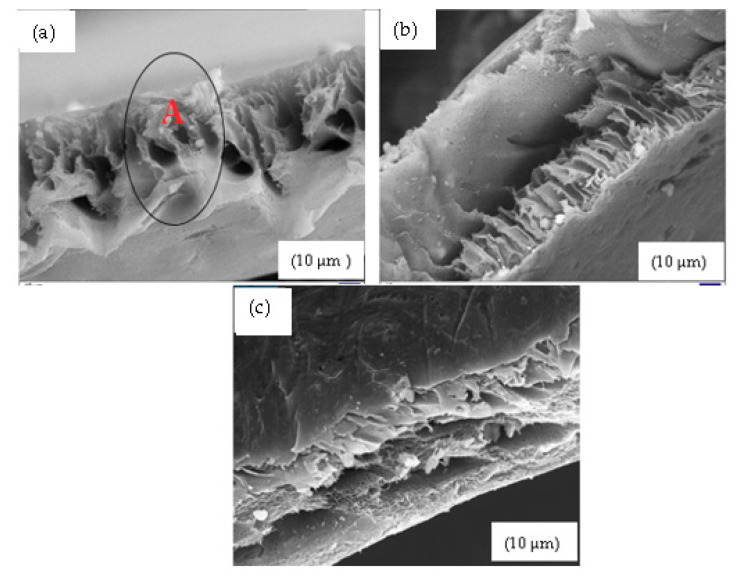
Cross-sectional view of: (**a**) 10% SSOD/Psf, (**b**) 10% fSSOD/Psf, and (**c**) 10% fSSOD/Psf/PVA.

**Figure 7 membranes-11-00315-f007:**
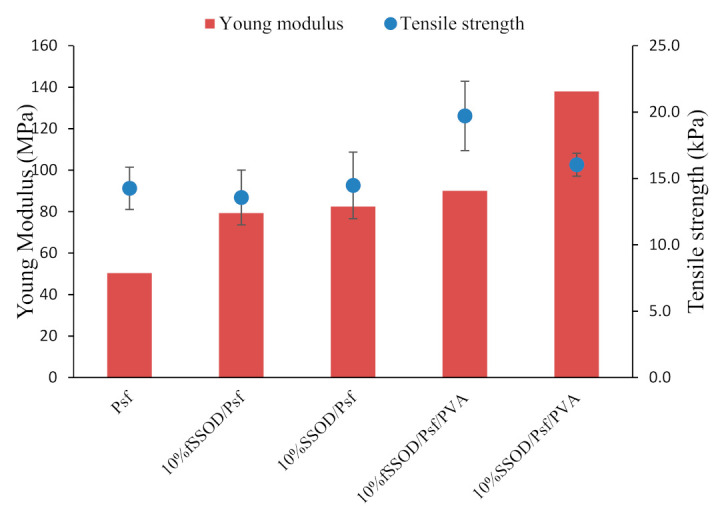
Comparison of the mechanical strength of the membranes.

**Figure 8 membranes-11-00315-f008:**
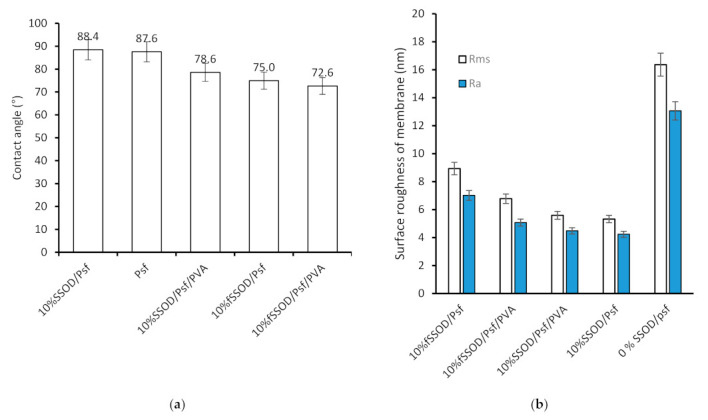
(**a**) Contact angle and (**b**) surface roughness of the fabricated membranes.

**Figure 9 membranes-11-00315-f009:**
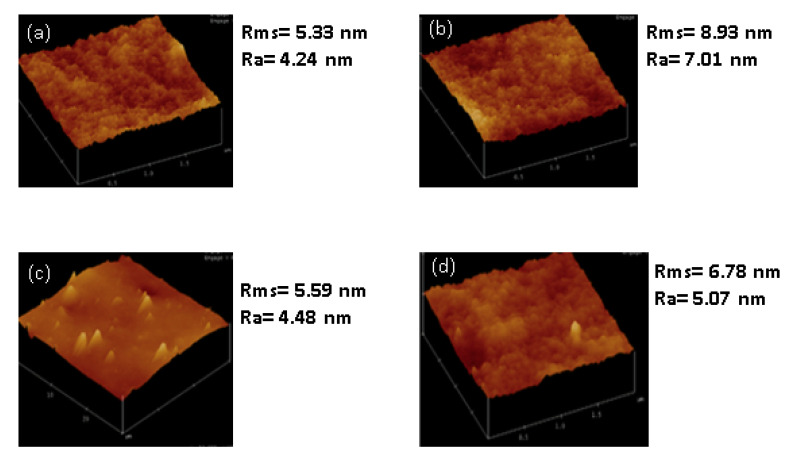
AFM images of: (**a**) 10% SSOD/Psf, (**b**) 10% fSSOD/Psf, (**c**) 10% SSOD/Psf/PVA, and (**d**) 10% fSSOD/Psf/PVA evaluated at an area of 2 × 2 µm^2^.

**Figure 10 membranes-11-00315-f010:**
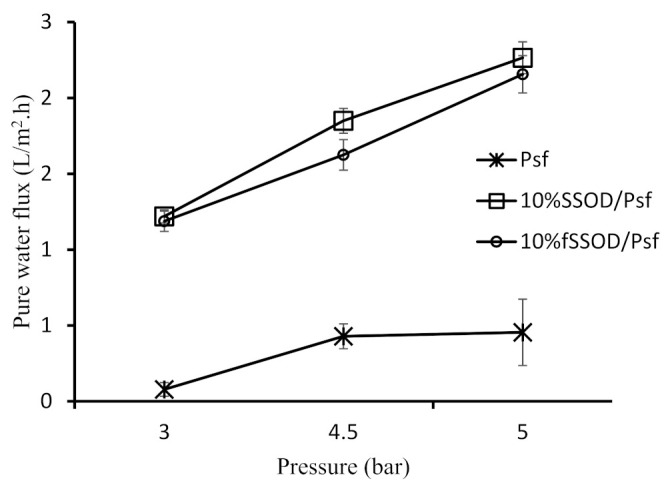
Effect of functionalization of nanoparticles on the pure water flux of the membranes. Experimental conditions: temperature: 25 °C and pressure: 3–5 bar.

**Figure 11 membranes-11-00315-f011:**
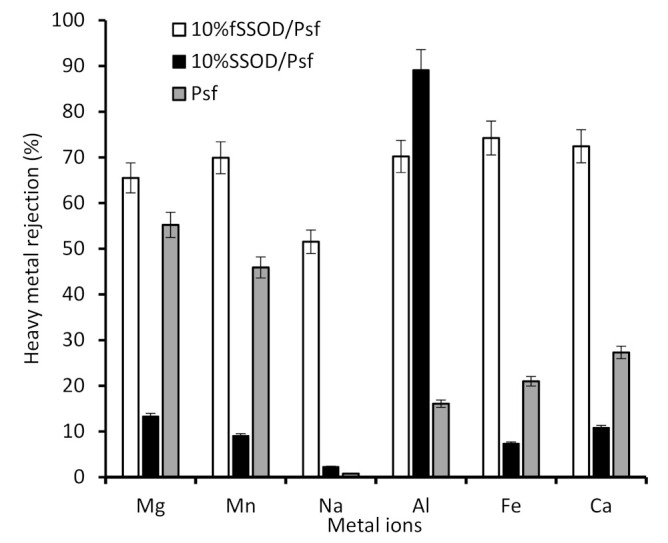
Heavy metal rejection from fSSOD and SSOD-loaded fabricated membranes.

**Figure 12 membranes-11-00315-f012:**
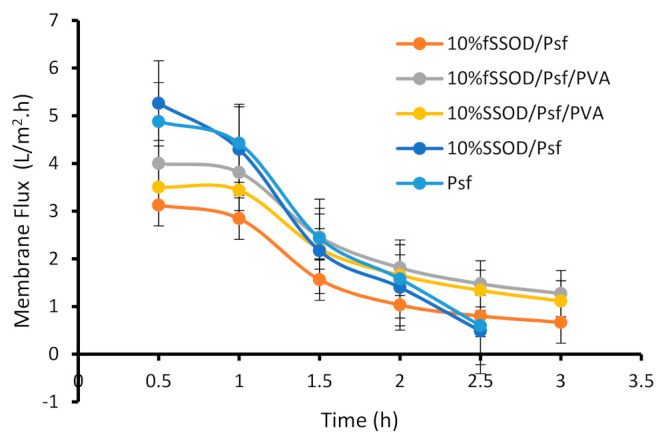
Effect of functionalization and PVA-coating on fouling property of the fabricated membranes. Experimental conditions: temperature: 25 °C and TMP: 4 bar.

**Figure 13 membranes-11-00315-f013:**
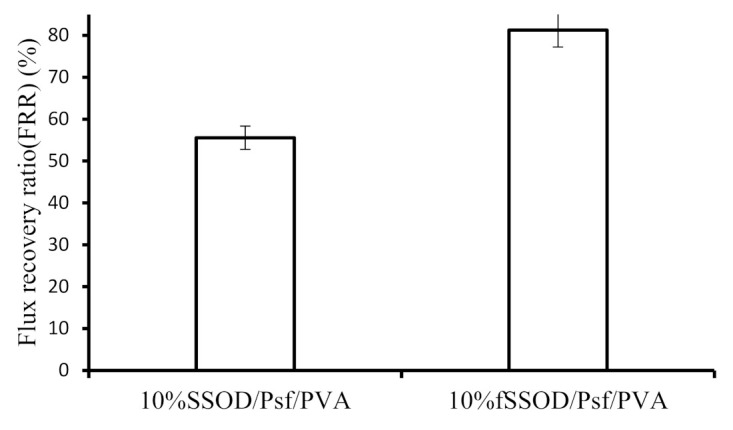
Flux recovery ratio of 10% SSOD/Psf/PVA and 10% fSSOD/Psf/PVA membranes. Experimental conditions: temperature: 25 °C and pressure 4 bar.

**Table 1 membranes-11-00315-t001:** Metal concentration of synthetic AMD and the atomic absorption spectroscopy operating parameters.

Cation	Salt	Concentration (mg/L) (Bell et al., 2001)	Sample Concentration (mg/L)	Lamp Current (mA)	Wavelenght (nm)	Flame Used (+Acetylene)
Mg^2+^	MgCl_2_	15.0	12.1	4.0	202.6	Air
Mn^2+^	MnCl_2_.4H_2_O	5.0	31.9	5.0	321.7	Air
Na^2+^	NaOH pellets	688.0	624.8	5.0	330.3	Air
Al^3+^	Al(NO_3_)_3_	84.0	89.1	10.0	237.3	N_2_O
Fe^3+^	Fe(NO_3_)_3_·9H_2_O	111.0	100.7	5.0	392.0	Air
Ca^2+^	Ca_2_OH_2_	41.0	59.5	10.0	239.9	N_2_O
SO_4_^2-^	NaSO_4_	1108.0	879.7	-	-	-

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
