# Peer review of "Effect of Silica Sodalite Functionalization and PVA Coating on Performance of Sodalite Infused PSF Membrane during Treatment of Acid Mine Drainage"

_membranes, 2021, doi:10.3390/membranes11050315_

Round 1

Reviewer 1 Report

This work investigates the effect of silica sodalite and coating on the performance of PSF membrane for the treatment of acid mine drainage. The concept is good but the design of the experiment can be further improved. Below are some suggestions on how this work can be improved:

1) It should be made clear that this is a NF membrane and the synthesis follow an NF recipe.

2) The polymer solution should be degassed before casting. Degassing can be done by leaving the polymer solution overnight or by ultrasonication (without the magnetic stirrer bar).

3) The writing will need to be improved. There are uncommon styles used in the discussion. E.g. "there is an understanding (line 78), is said (Line 232, 235) etc."

4) "Dr Blade" should read doctor blade or casting knife.

5) "#memcon" (Line 178) should read Memcon

6) Line 181: The thickness of the membrane should be replaced with the diameter of the actual membrane.

7) Line 188: PH should read pH

8) Line 189: This should be a separate section as it is about flux measurement and not the synthesis of AMD solution.

9) Precompaction of membrane before measuring the flux is missing.

10) Table 1: AA is not defined in the main text.

11) Figure 1: The scale bar of the two SEM images are not the same. The same scale bar should be provided to ease comparison. Also, it appeared that the SSOD nanoparticles are not well dispersed. Agglomeration of samples should be minimized.

12) Figure 2: FTIR figure needs to be improved. Y axis should be labelled and wavenumber should decrease from left to right (3900 to 400 cm-1).

13) The successful synthesis of SSOD cannot be confirmed. Authors need to provide evidence that SSOD and fSSOD are successfully synthesised. 

14) Figure 4: The pore size should be visible from the top surface of the membranes. To investigate the effect of loading of nanoparticles on the morphology of the membrane- SEM images of the top and bottom surface should be provided. SEM images of the control membrane is missing.

15) Figure 5: Typo: "Tensile strenght" should read "Tensie strength". "KPa" should read "kPa". Error bar is missing. How many repeats did the author perform on each sample?

16) Figure 6: Error bar is missing. Surface roughness of control sample is missing.

17) Line 348: The discussion on the membrane surface roughness is misleading. AFM was performed on membrane sample before fouling test. Any changes to the surface roughness cannot be due to fouling agents.

18) Figure 7: AFM of PVA coated membrane and control membrane is missing.

19) Line 374: Discussion on increase in pore size caused by nanoparticles can be improved. There are multiple factors to increased flux. The effect of loading of nanoparticle on the pore size, pore size distribution, morphology and hydrophilicity should be discussed. All these parameters affect the water flux of the membrane.

20) Figure 8: The pure water flux appeared to be very low for NF membranes. Authors should report permeance (LMH/bar) instead of pure water flux (LMH) in Figure 8.

21) Figure 9: Font size of y axis should be checked. Legend should be checked. "10fSSOD" should read "10%fSSOD"?

22) Line 411: The SEM image of the top surface of the membrane should be provided to confirm that the SSOD is on the top surface.

23) Figure 10: It is not clear at what pressure the flux was measured.

24) Figure 11: Control membrane is missing. The membranes in this figure are not the same as the other figure. In Fig 11. PVA coated membranes are reported. However, Fig 8 and Fig 9 report membranes without PVA. The same membranes should be reported and compared.

Reviewer 2 Report

The paper reports the synthesis of PSF membranes combined with SSOD and fSSOD nanoparticles. The membranes were characterized by various methods. Performance tests were carried out using synthetic wastewater. The authors claimed that the prepared membranes were promising to treat acid mine drainage. 

Although the paper is meaningful, its originality is relatively weak. Moreover, there is no breakthrough from this work. Instead, only a marginal progress was done. Besides, there are several issues to be addressed first. Accordingly, it is not recommended for publication in its current form.

The specific comment are as follows:

  1. PSF has a lot of disadvantages. Is it meaningful to use it for synthesizing new membranes?
  2. There is no error bars in the plots. Accordingly, the reproducibility cannot be confirmed. Besides, there is no statistical analysis on the experimental data.
  3. The current figure style is not suitable for journal publication. Moreover, the figure captions do not contain sufficient information. 
  4. The membrane performance evaluation is insufficient. More experiments should be done. Moreover, they should be multiplicated.
  5. The water flux values reported in this paper is too low and thus the prepared membranes seem to be useless for practical application. Did the authors compare their membranes with commercial membranes?
  6. One of the critical issues on nanocomposite membranes is their long-term stability. Without confirming it, the results are not meaningful at all. Accordingly, more experiments should be done to demonstrate the long-term stability and reusability. 
  7. In Fig.11, did the authors confirmed that the rejection values were the same as the those before the experiments? 

Round 2

Reviewer 1 Report

A few more suggestions are provided below.

1) Fig 1. Please remove 'at 30.00 kx magnification' as this is not true indicator of the actual size. Please include a scale bar.

2) There are two Fig. 2 in the manuscript. Please check figure numbering.

3) The resolution of FTIR spectra is very poor.

4) Fig. 4. Please include a scale bar and remove the 2.0 kx magnification. The pores for the membrane in Fig. 4 a is very large, even for an ultrafiltration membrane. Please confirm if the image is the top surface or the bottom surface of the membrane. Please ensure that the top side of the membranes are compared.

5) Fig. 9. The heavy meal rejection is not convincing. Based on the SEM image provided, the pore sizes are in the micron range which make it a lose UF or MF membrane.

6) What are the size of the heavy metal rejected? This will also tell us if the membranes are truly NF range.

Author Response

Thank you for the comments. Please find attached the document containing responses to your comments. 
